# Computed Tomography Does Not Improve Intra- and Interobserver Agreement of Hertel Radiographic Prognostic Criteria

**DOI:** 10.3390/medicina58101489

**Published:** 2022-10-19

**Authors:** Paulo Ottoni di Tullio, Vincenzo Giordano, William Dias Belangero, Robinson Esteves Pires, Felipe Serrão de Souza, Pedro José Labronici, Caio Zamboni, Felipe Malzac, Paulo Santoro Belangero, Roberto Yukio Ikemoto, Sergio Rowinski, Hilton Augusto Koch

**Affiliations:** 1Serviço de Ortopedia e Traumatologia Prof. Nova Monteiro—Hospital Municipal Miguel Couto, Rio de Janeiro 22430-160, Brazil; 2Departamento de Ortopedia, Reumatologia e Traumatologia—Universidade Estadual de Campinas (UNICAMP), Campinas 13083-970, Brazil; 3Departamento do Aparelho Locomotor—Universidade Federal de Minas Gerais (UFMG), Belo Horizonte 31270-901, Brazil; 4Departamento de Cirurgia Geral e Especializada—Universidade Federal Fluminense (UFF), Niteroi 24220-900, Brazil; 5Departamento de Ortopedia—Santa Casa de São Paulo, São Paulo 01221-020, Brazil; 6Departamento de Ortopedia e Traumatologia—Escola Paulista de Medicina, Universidade Federal de São Paulo (UNIFESP), São Paulo 04021-001, Brazil; 7Grupo de Ombro e Cotovelo—Faculdade de Medicina do ABC, Santo André 09060-870, Brazil; 8SUORT—Clínica Integrada, São Paulo 01258-010, Brazil; 9Departamento de Radiologia—Universidade Federal do Rio de Janeiro (UFRJ), Rio de Janeiro 21941-901, Brazil

**Keywords:** proximal humerus fracture, humerus head necrosis, avascular necrosis risk factors, posttraumatic avascular necrosis, Hertel criteria

## Abstract

*Background and Objectives*: Proximal humerus fractures are the second most frequent site of avascular necrosis (AVN), occurring in up to 16% of cases. The Hertel criteria have been used as a reference for the prediction of humerus head ischemia. However, these are based solely on the use of radiographs, which can make interpretation extremely difficult due to several reasons, such as the overlapping fragments, severity of the injury, and noncompliant acute pain patients. The objectives of the study were to evaluate the role of computed tomography (CT) in the interpretation of the Hertel criteria and to evaluate the intra- and interobserver agreement of orthopedic surgeons, comparing their area of expertise. *Materials and Methods*: The radiographs and CT scans of 20 skeletally mature patients who had fractures of the proximal humerus were converted to jpeg and mov, respectively. All images were evaluated by eight orthopedic surgeons (four trauma surgeons and four shoulder surgeons) in two different occasions. The intra- and interobserver agreement was assessed by using the Kappa coefficient. The level of significance was 5%. *Results*: There was a weak-to-moderate intraobserver agreement (κ < 0.59) for all examiners. Only the medial metaphyseal hinge greater than 2 mm was identified by 87.5% of evaluators both in the radiographic and CT examinations in the two rounds of the study (*p* < 0.05). There was no significant interobserver agreement (κ < 0.19), as it occurred only in some moments of the second round of evaluation. *Conclusions*: The prognostic criteria for humeral head ischemia evaluated in this study showed weak intra- and interobserver agreement in both the radiographic and tomographic evaluation. CT did not help surgeons in the primary interpretation of Hertel prognostic criteria used in this study when compared to the radiographic examination.

## 1. Introduction

Fractures of the proximal humerus represent 4 to 5% of all fractures, being the third most common in the human body [1]. Approximately 85% of patients can be treated non-surgically, particularly older patients with fractures involving the surgical neck of the humerus [2]. However, fractures considered more complicated require surgical treatment, especially in younger patients. Indeed, about 13 to 16% of all fractures of the proximal humerus are in three, four, or more parts, including the humerus head, greater tuberosity, lesser tuberosity, and shaft, and present unacceptable displacements [3]. Although a good functional outcome has been reported both after non-surgical treatment and after internal fixation [4], several postoperative complications have been described as a result of proximal humerus fractures, including avascular necrosis (AVN) of the humeral head [5].

Regardless of the type of treatment, proximal humerus fractures are the second most frequent site of AVN, occurring in up to 16% of cases [6]. The main risk factors for AVN are a greater number of fragments, head-split fracture, short segment of the humeral calcar, rupture of the medial hinge, displaced tuberosities, glenohumeral fracture/dislocation, and significant angular displacement of the head [7,8]. Other factors, such as the surgical approach and poor anatomical reduction, have also been implicated as risk factors [8]. Hertel et al. [7] radiographically evaluated 100 fractures of the proximal humerus in 98 patients who underwent internal fixation over a period of four years. These authors observed that good predictors of ischemia were posteromedial metaphyseal extension of the head less than 8.0 mm, medial hinge rupture (>2.0 mm), and fracture patterns involving the anatomical neck. The combination of these three factors led to a positive predictive value of 97% for the development of AVN of the humeral head.

Shortly after its publication, other studies from the same group observed high intra- and interobserver reliability of the Hertel classification, providing a more adequate description of proximal humerus fractures compared to the systems described by the Neer system and the AO group (Arbeitsgmenischaft für Osteosynthesefragen) [9,10]. However, there are some potential confounding factors in the study by Hertel et al. [7], such as the use of radiographs alone and the adoption of deltopectoral approach in all surgical procedures. In particular, the use of radiographs can make interpretation extremely difficult due to the overlapping fragments, severity of the injury, and noncompliant acute pain patients, making it impossible to perform all the shoulder standard views and not allowing substantial agreement [11].

We hypothesized that the adoption of computed tomography (CT) images, including three-dimensional reconstruction (3D CT), to radiographs of the proximal humerus in patients with fractures increases the reproducibility of the Hertel criteria, thus improving intra- and interobserver agreement among orthopedic surgeons. The objective of the study was to evaluate the role of CT in the interpretation of the Hertel criteria, using the radiographic evaluation of the same cases as a standard to identify the criteria described by these authors as prognostics for AVN of the humeral head.

## 2. Materials and Methods

Radiographs and CT scans of 20 skeletally mature patients who had fractures of the proximal humerus treated at the Institution between January and December 2020 were selected. Patients were chosen at random. We included patients older than 18 years, with a confirmed diagnosis of proximal humerus fractures, and who signed the Informed Consent form. We excluded patients with missing demographic data; inaccurate imaging exams; and presenting a pathological fracture or previous fracture history, surgical history, congenital deformities, or degenerative changes at the same region.

Patients’ privacy and security during the acquisition, storage, and transmission of their medical information were protected. The identity of the patients was not revealed. All patients were surgically treated and had standard preoperative radiographic and CT studies. The age ranged from 52 to 70 years, with a mean of 59.4 (SD ± 6.21882) years. Fifteen (75%) patients were female, and the right side was affected in 12 (60%) cases. The main injury mechanism was a fall to the ground in 17 (85.0%) cases (Table 1). The imaging evaluation comprised a true AP (Grashey), scapular Y, and axillary view [12]. In patients with limited abduction of the glenohumeral joint due to pain or joint incongruity, the modified axillary view was performed [13]. The CT evaluation comprised 5 mm axial, coronal, and sagittal slices and a 3D reconstruction. Both radiographic and tomographic images were kept anonymous for the purpose of this study.

Fractures were classified radiographically, according to the Neer system [12,14]. In addition, three of the Hertel criteria were evaluated on radiographs and on CT (Figure 1). All images were initially evaluated independently by two consultants who were fellowship-trained in orthopedic trauma and with more than 10 years of experience. In cases of disagreement, a senior consultant (with more than 20 years of experience in orthopedic trauma) evaluated the images. There were only 2 cases of disagreement, Cases 10 and 11. Both were further classified after the third evaluator as Neer 3-parts (head, greater tuberosity, and shaft), with medial metaphyseal extension < 8 mm observed on both the X-ray and CT. The standard (namely STANDARD) was considered when an absolute agreement was encountered between at least two evaluators.

The Standards for Reporting of Diagnostic Accuracy (STARD) questionnaire was used to determine surgeons’ perception of their accuracy in identifying each of the three criteria [15,16]. The selection was evidence-based whenever possible; therefore, the “inconclusive” category was added. Thus, both the evaluators defined as STANDARD and the respondents invited to participate in the study were asked to answer whether each of these three criteria was “present”, “absent” or “inconclusive” (cannot be evaluated) in the radiographic and tomographic examinations.

The radiographic images of the 20 cases were extracted from the DICOM (Digital Imaging and Communications in Medicine) disk, which was developed by the American College of Radiology (ACR) and the National Electrical Manufacturers Association (NEMA—Arlington, VA, USA); converted to jpeg (Joint Photographic Experts Group) with 1000 × 1000 pixels and 300 DPI resolution; and stored case by case in individual folders, from 01 to 20 [17]. In the same way, the tomographic images of the 20 cases were extracted from the DICOM disk, recorded as individual frames, and saved in the mov (Multimedia Container File) format, a file name extension for the QuickTime multimedia file format (Apple Inc., Cupertino, CA, USA), with a resolution of 1228 × 657 ppi [18]. Figure 2, Figure 3 and Figure 4 illustrate some of the cases used in the study.

The images were sent by email to eight board-certified orthopedic surgeons with more than 10 years of experience in the treatment of proximal humerus fractures (Observers 1–8), in 20 individual folders, along with instructions on how to respond (Appendix B). All possible data that could identify the patients, such as name, initials, and date of birth, were removed from the exams so that their confidentiality was fully preserved. Four surgeons with fellowship in orthopedic trauma (namely TRAUMA 1 to TRAUMA 4) and four surgeons with fellowship in shoulder and elbow (namely, SHOULDER 1 to SHOULDER 4) were invited. Surgeons were asked to determine whether each of the Hertel prognostic criteria used in the study was “present”, “absent”, or “inconclusive” on radiographic and CT imaging. Respondents received the illustration of the prognostic criteria used in the study. They were asked to start with the radiographic images and not store images on their computers. Images were evaluated on two occasions, separated by 15 days—Round 1 (R1) and Round 2 (R2). For R2, the 20 folders were randomized and sent back to respondents by email, with the same guidelines as in R1.

The inferential analysis consisted of the Kappa coefficient (κ) to assess intraobserver agreement in the criteria (positive, negative, and inconclusive) according to radiographic and tomographic images [19]. Landis and Koch [19] suggest the following interpretation: κ < 0.19—no/very poor agreement; 0.20 < κ < 0.39—weak agreement; 0.40 < κ < 0.59—moderate agreement; 0.60 < κ < 0.79—good agreement; and κ ≥ 0.80—very good (excellent) agreement. The interobserver agreement was defined by the level of concordance (positive/negative/inconclusive) in the cases divided by the total number of cases (*n* = 20) and multiplied by 100 (% of concordant cases). The interobserver agreement of the eight surgeons and the STANDARD was provided by four percentages of concordant cases (positive/negative/inconclusive/general). The significance level adopted was 5%. Statistical analysis was performed by using SPSS version 26 software (IBM, New York, NY, USA).

## 3. Results

### 3.1. Intraobserver Agreement

There was a weak-to-moderate intraobserver agreement for all examiners. For analytical consistency, the percentage of agreement was judged useful only when the Kappa coefficient was significant (Appendix A). Only the medial metaphyseal hinge greater than 2 mm (Criterion B) was identified by all evaluators both in the radiographic and CT examinations in the two rounds of the study (*p* < 0.05), except by the evaluator SHOULDER 4. Both Criteria A and C were identified significantly less frequently in the two rounds of the study, with no difference between radiographic and tomographic evaluations (*p* > 0.05). The SHOULDER 4 evaluator showed no intraobserver agreement for any of the criteria evaluated in R1 and R2.

### 3.2. Interobserver Agreement

The percentage of “inconclusive” concordant cases by CT in R1 and R2 was not processed due to improper operation (division by zero), as there were no “inconclusive” cases by CT observed according to the STANDARD. There was practically no significant agreement with the STANDARD, as it occurred only in some moments of R2. Interobserver agreement was seen as weak in both R1 and R2. Despite this, it was observed that there is strong agreement between the imaging methods used in the study. There was significant agreement at a level of 5% for the criteria between the radiographic and tomographic exams of moderate-to-strong degree for almost all evaluators, regardless of the moment (R1 and R2). However, it was observed that the radiographic evaluation presented a higher number of the “inconclusive” category than the CT, regardless of the criterion (Criteria A, B, and C) and the moment (R1 and R2), although there was no statistical significance. Appendix A provide the percentage of concordant cases in R1 and R2, respectively.

## 4. Discussion

Overall, it was observed that intra- and interobserver agreement was weak in both R1 and R2. Analyzing individually each of the prognostic criteria used in this study, it was observed that only the presence of a medial metaphyseal hinge greater than 2 mm (Criterion B) was identified by all evaluators both in the radiographic examination and in the CT in the two rounds of the study as statistically significant, meaning that there is no evidence supporting our hypothesis. There was practically no significant agreement with the STANDARD, occurring only in some moments of R2. Nevertheless, there was a strong agreement between the imaging methods used in the study for almost all the evaluators, regardless of the moment (R1 and R2), with the CT evaluation showing a lower number of the “inconclusive” responses than the radiography evaluation, regardless of the criterion (A, B, and C) and moment (R1 and R2).

It is known that posttraumatic osteonecrosis of the proximal humerus is the most common complication after fractures in this anatomic region, occurring in up to 16% of patients, and represents a problem for both the patient and the surgeon [6,8,20]. In this context, surgeons must be able to assess the risk factors for AVN of the humeral head to influence not only decision-making, but also to guide them regarding the prognosis in relation to this complication [13,14,21]. Hertel et al. [7] described some predictors of humeral head ischemia after fracture; however, little is discussed about the difficulty of interpreting these criteria in the presence of numerous factors that potentially make it difficult to evaluate the radiographic examination.

The guidelines recommend that radiographic screening should be the first imaging investigation in the emergency department [22]. However, a high rate of suboptimal shoulder radiographs has been identified, particularly in AP and axillary views, resulting in increased workload, increased radiation to patients, inconvenience and decreased patient satisfaction, and increased risk of incorrect or missed diagnoses [23]. The variability in interpretation and the questioned reliability of this test have led some authors to independently assess its effectiveness. Martínez-Sola et al. [24] found a low-to-moderate degree of interobserver agreement, using a single AP radiograph of the shoulder, denoting the difficulty of interpretation by orthopedic surgeons of various levels of experience of some of the currently most used classifications for fractures of the proximal humerus. Likewise, Iordens et al. [25] found weak intra- and interobserver agreement by using radiographs of the proximal end of the humerus for the Hertel criteria. Most likely because of this difficulty in radiographic evaluation, which can be understood as a confusing diagnostic and prognostic factor, some authors observed that the Hertel criteria were not sufficient to determine a greater chance of progression to osteonecrosis of the humeral head [11]. Analyzing specifically the three prognostic criteria evaluated in this study, we can see that the presence of a medial metaphyseal hinge greater than 2 mm (Criterion B) was identified in a statistically significant way by all evaluators, except the evaluator SHOULDER 4. Interestingly, from this evaluator, we observed that there was an almost symmetrical pattern of inversion of his responses between R1 and R2, thus leading us to believe that an attention bias may have occurred during the responses. Thereby, what was considered to be a bad prognostic factor in R1 was considered to be a good prognostic factor in R2, and vice versa.

It is known that there is a strong correlation between AVN of the humeral head and medial metaphyseal hinge [7,20,26]. Hertel et al. [7] found an accuracy of 0.79 for ischemia when the medial hinge was interrupted by more than 2 mm. Solberg et al. [20] noted the occurrence of osteonecrosis in all patients in whom the medial hinge was initially less than 2 mm in length. However, these authors [20] were unable to identify whether medial hinge extension or a history of dislocation was the specific cause of osteonecrosis. Humeral head AVN has been reported in up to 33% of patients, and late surgery (>48 h) appears to be an important prognostic factor for ischemia [27]. However, while a history of dislocation associated with a proximal humerus fracture could be expected to lead to osteonecrosis from capsular injury and direct vascular damage, this is not fully supported in the literature [8,26,28,29,30,31]. Interestingly and antagonistically, Neviaser et al. [29], in a series of 34 patients treated with open reduction and internal fixation of fractures of the proximal humerus, showed that the posteromedial hinge length was not an accurate parameter to predict the risk of osteonecrosis of the humeral head, although it has been shown to be useful for surgical planning, especially as it involves an important support region (calcar) for internal fixation, greatly reducing the incidence of cut-out and/or cut-through [31].

The prognostic ability of the other two criteria evaluated in this study (medial metaphyseal extension in the cephalic component less than 8 mm—Criterion A; and fracture by partition of the head—Criterion C) is even more controversial and, probably, is mainly due to the inability to fully assess the proximal humerus morphology in a fractured segment with radiographs alone, as pointed out before. In this sense, the use of CT gains importance, as it allows the surgeon and the radiologist to perform the reconstruction of the proximal humerus in different planes, with slices of small sizes and in a three-dimensional perspective; however, in the light of current knowledge, there is no evidence of that 3D CT is superior to 2D CT [32]. Campochiaro et al. [11] suggested that all fractures involving the calcar should be studied with CT, as an accurate assessment of the fracture in three planes is necessary. In this study, only the presence of a medial metaphyseal hinge greater than 2 mm (Criterion B) was identified in a statistically significant way by all evaluators in the CT, except for the SHOULDER 4 evaluator; meanwhile, the other two criteria showed little uniformity between evaluators. There was a trend toward greater identification of Criteria A and C by TRAUMA surgeons on CT; however, this was not statistically significant.

This study has some limitations. Although not evaluated in the study, there are other factors that may represent risk factors for AVN of the humeral head—such as the surgical approach and the quality of reduction—and generate confusion in its epidemiological estimate, which is illustrated by the widely varying rates of posttraumatic osteonecrosis between studies [8]. In addition, the reported rate of humeral head AVN depends, among other risk factors already mentioned, on the duration of patient follow-up, the intensity of the resulting symptoms, and the imaging methods used to diagnose its presence [4,5,6,33,34]. In addition, the study design did not include the follow-up of patients, thus making it impossible to observe which patients developed AVN of the humeral head and to correlate this complication with the findings of both the evaluators and the STANDARD. This information would be important to assess the accuracy of the prognostic criteria investigated in the study and to understand the lack of interobserver agreement with the STANDARD. Moreover, the use of images in JPEG format is biased due to zoom or brightness adjustments compared to dedicated software for viewing radiographs. Furthermore, more dedicated software for the MOV file would allow all CT planes to be viewed simultaneously, and this would theoretically increase accuracy and orientation when viewing a CT scan. The adoption of these measures could theoretically increase intraobserver agreement, which can be tested in future studies. Finally, the objective was to evaluate the role of both two- and three-dimensional CT in the interpretation of the three criteria described by Hertel, using the radiographic evaluation of the same cases as a standard. Unlike other studies [25,35,36,37,38], our study did not seek to assess whether there was a difference between 2D and 3D tomography. In the current study, we showed some usefulness for both the 2D and 3D tomography to help surgeons in the primary interpretation of Hertel prognostic criteria, with moderate intraobserver agreement, without, however, statistical difference when compared to the radiographic examination. Recent studies have shown that the use of other imaging modalities, such as segmented 3D CT [39] and 3D printed models [40,41], adds value to the understanding of the morphology of the proximal humerus fracture, which was not evaluated in the current study.

Some strengths of the study deserve to be highlighted. The main one was to show that none of Hertel prognostic criteria evaluated in this study present interobserver agreement, regardless of whether the assessment is performed by using radiographs, CT, or both. Other authors have already observed the same, including employing more modern imaging methods than those used in this study [11,25,39,41]. The main limitation of the study by Hertel et al. [7] was the assessment of the fracture pattern by a single observer, which obviously lacks objectivity and reproducibility. The prognostic value of the Hertel criteria for decision-making has been questioned in particular because it has been reported that the humeral head can survive even in the initial absence of proximal flow from the anterior and posterior branches of the humeral circumflex artery [42]. Thus, our findings (and those of others) indicate that Hertel prognostic predictors should be used cautiously and sparingly when defining the eventual risk of AVN of the humeral head due to their low interobserver agreement.

## 5. Conclusions

The prognostic criteria for humeral head ischemia evaluated in this study showed weak intra- and interobserver agreement in both radiographic and tomographic evaluation. CT did not help surgeons in the primary interpretation of Hertel prognostic criteria used in this study when compared to radiographic examination.

## Figures and Tables

**Figure 1 medicina-58-01489-f001:**
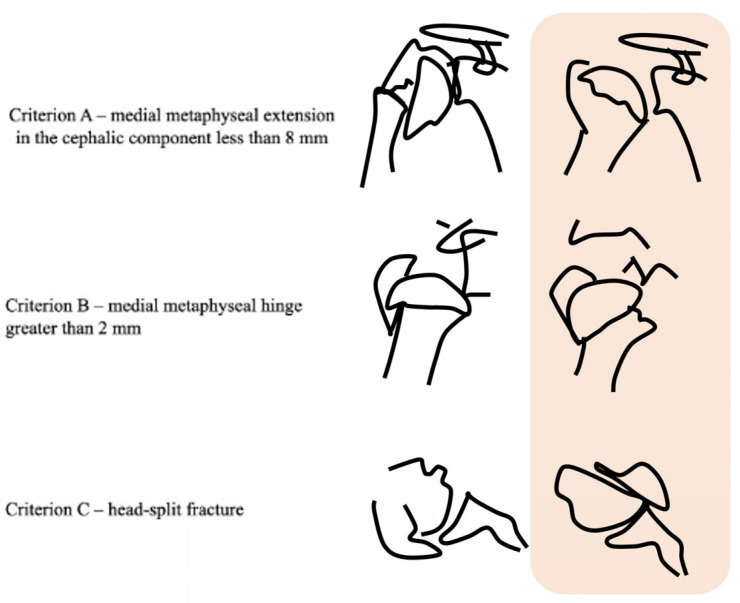
The three Hertel criteria adopted in the study. Criterion A represents the medial metaphyseal extension < 8 mm; Criterion B represents the medial metaphyseal hinge < 2 mm; and Criterion C represents the humerus head-split fracture. (Image produced on computer and from the author’s personal archive VG).

**Figure 2 medicina-58-01489-f002:**
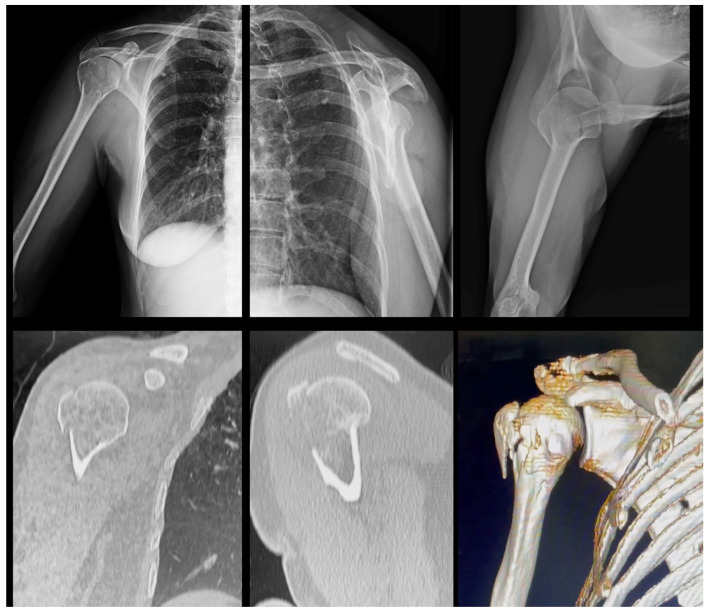
Radiographic and CT images of a 53-year-old female patient with a right proximal humerus fracture (Case 2). Note that none of Hertel’s prognostic criteria is present in either of the two imaging exams. The fracture was classified as Neer 2-parts (greater tuberosity).

**Figure 3 medicina-58-01489-f003:**
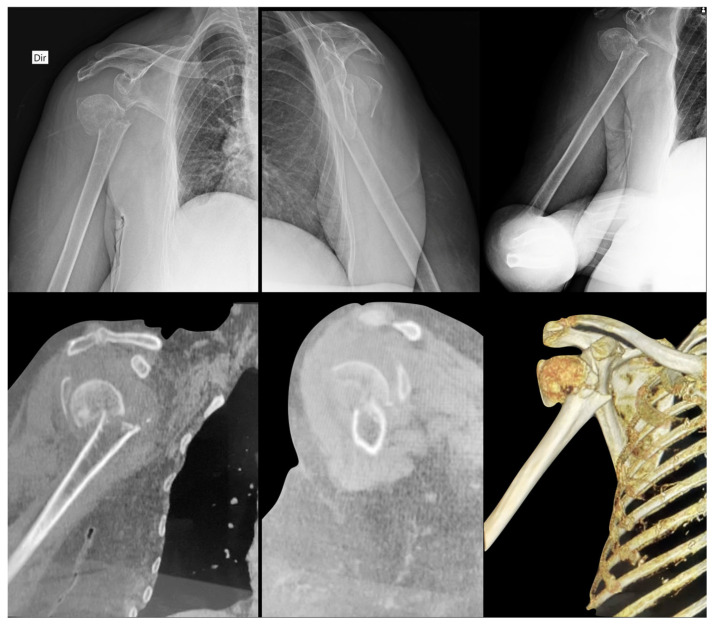
Radiographic and CT images of a 52-year-old female patient with a right proximal humerus fracture (Case 3). Note that the medial metaphyseal extension in the cephalic component less than 8 mm (Criterion A) and the medial metaphyseal hinge greater than 2 mm (Criterion B) are present and easily identified in both imaging exams. The fracture was classified as Neer 3-parts (head, greater tuberosity, and shaft).

**Figure 4 medicina-58-01489-f004:**
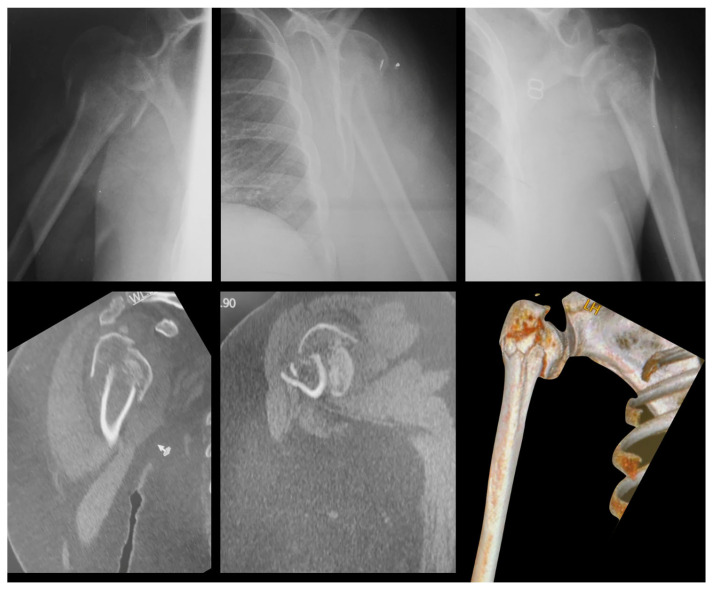
Radiographic and CT images of a 67-year-old female patient with a right proximal humerus fracture (Case 9). Note that it is possible to observe the split-head fracture of the head (Criterion C) only on CT, especially in 3D reconstruction. The fracture was classified as Neer 4-parts.

**Table 1 medicina-58-01489-t001:** Patient demographics and STANDARD assessment (*n* = 20).

Case	Age (Years)	Gender	Side	Mechanism of Trauma	Neer (Parts)	Hertel					
						X-ray C1	X-ray C2	X-ray C3	CT C1	CT C2	CT C3
**1**	**58**	**M**	**R**	**MCA**	III (H, GT, S)	P	A	A	P	A	A
2	53	F	R	Fall to the ground	II (GT)	A	A	A	A	A	A
3	52	F	R	Fall to the ground	III (H GT, S)	P	P	A	P	P	A
4	68	M	R	Fall to the ground	II (LT)	P	P	A	P	P	A
5	57	F	L	Fall to the ground	IV	A	A	I	A	A	A
6	65	F	L	Fall to the ground	IV	I	A	A	A	A	A
7	55	F	L	Fall to the ground	IV	A	A	I	A	A	P
8	64	F	R	Fall to the ground	IV	P	P	A	P	P	A
9	67	F	R	Fall to the ground	IV	A	A	I	A	A	P
10	59	M	L	MCA	III (H, GT, S)	P	A	A	P	A	A
11	53	M	R	MVA	III (H, GT, S)	P	A	A	P	A	A
12	56	F	L	Fall to the ground	II (H)	P	A	A	P	P	A
13	52	F	R	Running over	III (H, GT, S)	P	P	A	P	P	A
14	70	F	R	Fall to the ground	IV	P	A	A	P	A	A
15	66	F	L	Fall to the ground	II (LT)	P	P	A	P	P	A
16	58	F	R	Fall to the ground	III (H, GT, S)	I	A	A	A	A	A
17	54	F	L	Fall to the ground	II (LT)	P	P	A	A	A	A
18	69	M	R	Fall to the ground	II (H)	P	A	A	P	A	A
19	53	F	L	Fall to the ground	II (H)	A	A	A	A	A	A
20	59	F	R	Fall to the ground	III (H, GT, S)	A	A	A	A	A	A

Source: SOT-Nova, HMMC, 2022. Legends: M—male; F—female; L—left; R—right; II—two parts; III—three parts; IV—four or more parts; H—head; GT—greater tuberosity; LT—lesser tuberosity; S—shaft; X-ray—radiography; CT—computed tomography; C1—Criterion A; C2—Criterion B; C3—Criterion C; P—present; A—absent; I—inconclusive; MCA: motorcycle accident.

## Data Availability

Not applicable.

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
