# Peer review of "Computed Tomography Does Not Improve Intra- and Interobserver Agreement of Hertel Radiographic Prognostic Criteria"

_medicina, 2022, doi:10.3390/medicina58101489_

Round 1

Reviewer 1 Report

Title:

Ok.

Abstract:

Line 24: …recently?  ~20 years….

Edit this sentence please.

Hertel R, Hempfing A, Stiehler M, Leunig M. Predictors of humeral head ischemia after intracapsular fracture of the proximal humerus. J Shoulder Elbow Surg. 2004 Jul-Aug;13(4):427-33. doi: 10.1016/j.jse.2004.01.034. PMID: 15220884.

Line 26: “due to several reasons” this sentence is unclear. Please clarify or include in the background of the abstract what is the inter\intra observer rate to emphasize the rationale of your study.

 Introduction:

Line 50: please clarify what parts mean? Based on Neer’s classification? If so, insert reference, for the sake of the “average” reader.

 Line 62: see my previous comment regarding the term “recently”, also, you cited twice the reference [6] , consider editing the paragraph to avoid repetition.

 The intro lacks mentioning past attempts to solve the research problem, or address if this is the first study to evaluate this method.

 Methods:

Line 86-89: Move to result section.

Add inclusion\exclusion criteria for patients’ selection in this study.

Line 121:  Please elaborate regarding the view of images in JPEG format, as in this format there is a bias due to zoom or brightness adjustments compared with dedicated software to view radiographs.

Or consider adding this to limitations.

Elaborate more regarding the MOV file, as in a dedicated software, one can view all CT planes simultaneously which adds to the precision and orientation when viewing a CT scan. This in my opinion a major fault in this study. Please clarify.

 Results:

 Please add information (beside Table 1) regarding the case set of this study. How many cases were identified by the 2\3 independent evaluators before they were sent ? This information is important, did they identify the three criteria for all 20 patients? How did you choose to divide the cases? How the ratio between “normal” and “abnormal” cases were chosen?

Please clarify.

 Discussion:

Line 197-209: A repetition of the results section. Please edit. Start the discussion with the most important findings of the study.

Line 212: “huge problem”, please edit.

Line 215-223: “Hertel et al. [6] described some predictors of humeral head ischemia after fracture, being considered as good predictors the length of the metaphyseal extension in the head fragment, the integrity of the medial hinge, and the basic fracture pattern. These authors showed that the combination of all three factors led  to a positive predictive value of 97% for the development of humeral head ischemia. However, little is discussed about the difficulty of interpreting these criteria in the presence of numerous factors that potentially make it difficult to evaluate the radiographic examination, such as the severity of the injury, the action of muscle deforming forces, and the patient's pain during the imaging exam.”

 This is again a repetition of the same paragraph that is mentioned in the introduction.

The discussion should include the following:

1.Summarize the key findings in clear and concise language

2. Acknowledge when a hypothesis may be incorrect

3. Place your study within the context of previous studies

4. Discuss potential future research

5. Provide the reader with a “take-away” statement to end the manuscript

https://plos.org/resource/how-to-write-conclusions/

Line 248-263: I am not sure how this paragraph is relevant or adds to the discussion, please clarify or remove.

The following paragraph and this section of the discussion refers to the limitations and the disadvantages of the Hertel classification. In my opinion, this calls into question the importance of the current research, does the classification even have value? And if not, then why investigate and examine the use and comparison between photographs and CT?

The discussion can be written differently, for example, try to explain that based on CT, different and useful conclusions can be drawn compared to using Hertel's criteria.

A major revision is needed.

Limitations:

Please see my previous comment regarding JPEG and MOV files.

Line 283: this limitation is not related to the present study purpose or metthods.

 Line 305: “Other authors had already observed the same, including employing more modern imaging methods than those used in this study….”

These studies should be included in the discussion in detail including their main findings.

Even in the introduction it is possible to cite or elaborate, because the introduction does not refer at all to past studies on the subject.

Figure 1: Is that original figure? If not, add permission number.

Reviewer 2 Report

Dear Authors, the topic regarding the prognostic criteria for humeral head ischemia  is estremely interesting first of all in relation of hertel criteria evaluation.

As regards the introduction from line 47 to line 55 i suggest to improve this period citing the following articles in which Authors reported the results of shoulder arthroplasty and  psycological evaluation in patients with humeral fratures:

-Plate vs reverse shoulder arthroplasty for proximal humeral fractures: The psychological health influence the choice of device?

Maccagnano G et al.

World Journal of Orthopedics Open Access Volume 13, Issue 3, Pages 297 - 30618 March 2022

In fact Authors analyzed the importance of psycological evaluation in patients with humeral fractures in relation to AVN and type of surgery . It will be interesting for readers to know that there is an increasing interesting in this field.

As regards the M&M and results the section is well described

As regards the discussion and conclusions, are balanced and well supported by analysis . Finally your conclusion is strong and important because you underlined for first time in bibliography that the ct scan couldn’t help the surgeon for the evaluation of hertel criteria

Round 2

Reviewer 1 Report

 I congratulate the authors for a well written article on the subject.

The authors addressed every issue raised and the manuscript in its current form has been improved.